# Self-stigmatization and treatment preferences: Measuring the impact of treatment labels on choices for depression medications

Juan Marcos Gonzalez Sepulveda [1]*, Michael Townsend[2], Heidi C. Waters[3], Maalak Brubaker[3], Matthew Wallace[1], Reed Johnson[1]

1 Duke Clinical Research Institute, Duke University School of Medicine Durham, North Carolina, United States of America, 2 LCSW Gateway Counseling Center, Smithtown, New York, United States of America, 3 Otsuka Pharmaceutical Development & Commercialization, Inc., Princeton, NJ, United States of America

* jm.gonzalez@duke.edu

**Data Availability Statement:** Data cannot be shared publicly because of ownership restrictions.

## Abstract

### Objective

To collect evidence on the possibility that patients with depression experience self-stigmatization based on label information for medications.

### Methods

We developed a discrete-choice experiment (DCE) survey instrument that asked respondents to make choices between hypothetical treatments for major depressive disorder (MDD). We also included treatment type (antidepressants versus antipsychotics) and approved indications for the medication. The choice questions mimicked the information presented in product inserts and required systematic tradeoffs between treatment efficacy, treatment type, and indication. We calculated how many patients were willing to forgo efficacy to avoid treatments with information associated with self-stigmatization, and how much efficacy they were willing to forgo. We also evaluated the impact of contextualizing the treatment information to reduce self-stigmatization by randomizing respondents who received additional context.

### Results

A total of 501 patients with MDD were recruited to complete the DCE survey. Respondents had well-defined preferences for treatment outcomes. Over 60% (63.4%) of respondents were found to be significantly affected by treatment indication. These respondents were willing to forgo about 2.5 percentage points in the chance of treatment efficacy to avoid treatments indicated for schizophrenia. We also find that some level of contextualization of the treatment details could help reduce the negative impact of treatment type and indications.

Data are available from the authors and/or from Lundbeck, LLC upon request for researchers who meet the criteria for access to confidential data. All data requests must be made to Dewilka Saleem Senior Manager, Global Scientific Communications dewilka.saleem@otsuka-us.com.

**Funding:** Funding/Support: This work was funded by Lundbeck, LLC and Otsuka Pharmaceutical Development & Commercialization, Inc. The funder supported design and conduct of the study. However, all decisions regarding the study design and conduct, as well as data analysis and manuscript preparation were made by researchers unaffiliated to the funder.

**Competing interests:** I have read the journal's policy and the authors of this manuscript have the following competing interests: HCW and MB are employees of Otsuka Pharmaceutical Development & Commercialization, Inc. JMGS, RJ, and MW are employees of Duke Clinical Research Institute, which received funding to conduct this study. MT received compensation from Otsuka Pharmaceutical Development & Commercialization, Inc. for his review of the protocol and interpretation of the data for this study. This does not alter our adherence to PLOS ONE policies on sharing data and materials

## Conclusions

Product-label treatment indication can potentially lead to patient self-stigmatization as shown by patients' avoidance of treatments that are also used to treat schizophrenia. While the effect appears to be relatively small, results suggests that the issue is likely pervasive.

## Introduction

Decisions to pursue therapies for major depressive disorder (MDD) should correspond to the clinical characteristics of the patient, the evidence available on the expected treatment-related outcomes, and the relative importance of those outcomes. However, clinical decision-making can be the result of a more complex process that extends beyond clinical aspects and into the decision maker's own biases. Patients with mental health conditions can be particularly vulnerable to the phenomenon of self-stigmatization, in which the patients experience harmful levels of negative self-image because of the social stigma associated with their disease [1,2].

While there is strong evidence that self-stigmatization among patients with mental health issues can lead to avoidance of medical attention by a professional [2,3] or to treatment nonadherence [4–7], it remains uncertain whether associations with conditions not directly experienced by the patients exacerbate self-stigmatization. Specifically, whether knowing that a treatment is commonly prescribed to patients with *other* stigmatized conditions leads to avoidance of the treatment. Self-stigmatization may occur, even when the patient knows he or she does not have the stigmatized health issues.

In the context of MDD, patients who experience a partial or inadequate response to antidepressants may be recommended to use adjunctive atypical antipsychotics. These therapies can offer significant improvements in depressive symptoms [8]. However, per the United States (US) Food and Drug Administration (FDA) requirements, the product prescribing information for these therapies contains details about drug class (i.e., atypical antipsychotics) and highlights the fact that these therapies are also used for the treatment of patients with schizophrenia or other psychiatric disorders [4]. Of particular concern is that negative associations with psychosis and schizophrenia may lead to aversion to adjunctive therapies due, at least in part, to patients' self-stigmatization and forgoing the benefits that antipsychotics can provide.

This study assessed whether preferences for treatments for MDD among patients with inadequately managed depression were consistent with self-stigmatization when considering atypical antipsychotic adjunctive medications. Specifically, we sought to quantify patients' willingness to forgo treatment efficacy to avoid product that included an indication for schizophrenia, all else equal. Furthermore, this study evaluated how alternative terms to describe these therapies and contextualization of the label information could change patients' reactions and decrease resistance towards possible treatment options, thereby providing evidence to inform prescribers' counseling strategies.

## Methods

We developed a discrete-choice experiment (DCE) survey instrument that asked respondents to make choices between hypothetical treatments for MDD to measure the relative importance of different aspects in the scenarios presented [9]. The treatments were described based on general attributes, and how the treatment performs under each attribute (attribute level). The

**Table 1. Items considered in the DCE questions.**

| Attribute | Attribute Levels |
|---|---|
| Chance of the medicine working well to treat depression symptoms | 10 out of 100 (10%)<br>15 out of 100 (15%)<br>18 out of 100 (18%)<br>• 20 out of 100 (20%) |
| Medication Type | • Antidepressant<br>• Atypical antipsychotic<br>• Serotonin-dopamine activity modulator |
| Indication | • Major depressive disorder<br>• Major depressive disorder and schizophrenia |
| Dosage and administration | • Daily oral tablet at home<br>• Monthly injection at a clinic |
| Weight gain | None<br>• Most patients experienced a 2% increase in weight<br>• Most patients experienced a 7% increase in weight |
| Akathisia | • None<br>• Most patients have experienced akathisia |

attributes were identified by a literature review of treatments for MDD not responsive to anti-depressants, product inserts for treatments for MDD not responsive to antidepressants, and discussions with clinical experts. Table 1 includes the full list of study attributes and the list of possible levels shown to respondents under each attribute.

We included two attributes potentially associated with self-stigmatization. First, medication type, which included antidepressants, atypical antipsychotics, and serotonin-dopamine activity modulator. The second attribute showed the approved indications for the medication. The levels for this attribute included only *MDD*, and *MDD and schizophrenia*.

To prevent eliciting a reaction from participants beyond those typically triggered by the product label, we mimicked the information available to patients through product inserts as we described the attributes. This approach resulted in definitions that were short and technical, not just for the attributes related to self-stigmatization, but for all attributes other than efficacy —which is not directly covered in the product insert. We also randomized additional information for the description of medication type. The additional information was intended to provide more context around the use of atypical antipsychotics to treat MDD mimicking the kind of information a patient could receive from physicians to alleviate problems with self-stigmatization. Respondents who were offered the additional information (context arm) were reminded explicitly that patients need not experience psychoses to benefit from antipsychotics. We also allowed respondents to click a link for additional information on akathisia (inability to remain still) crafted by the study team and recorded whether respondents clicked on the link.

Our focus on the product label extended to the presentation of the DCE questions (Fig 1). That meant presenting the levels within alternatives following the presentation of medication labels in product inserts. While we presented the information in the form of a product insert, our intention was not to address the impact of inserts per se, but the information about the product conveyed by regulatory approval in a particular indication or medication category. In that sense, the use of a layout that mimicked the product insert was meant to capture the way such information may be accessed by patients. As mentioned before, this question layout was chosen to avoid highlighting the treatment type and indication in a way that would lead patients to overly focus on this information. We also asked respondents to state what treatment

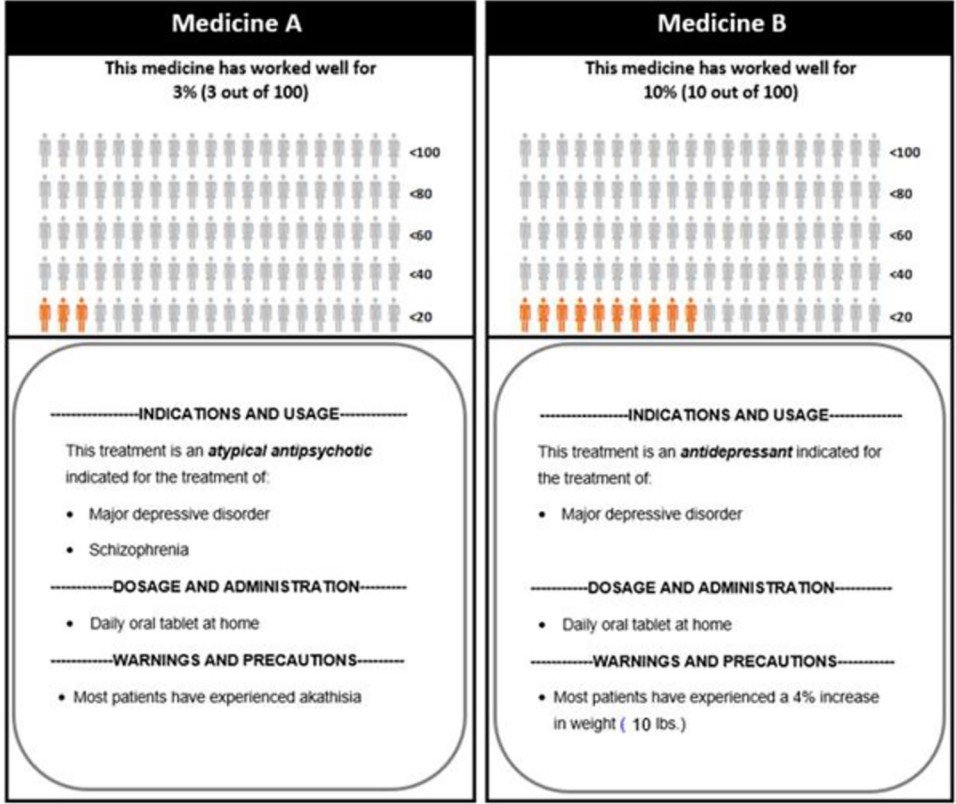

**Fig 1. Example DCE question.**

they would choose first, allowing them to consider the possibility of using the rejected treatment in the future. This format of the choice question is also consistent with the decision context for these patients.

The survey instrument was tested during 10 individual interviews with a convenience sample of patients with MDD. The interviews were conducted individually through videoconferencing and followed a semi-structured think-aloud protocol. During the 1-hour individual interviews we asked participants to read the survey out loud and to comment as needed throughout the document on things that seemed unclear or did not match their own experience with the disease or treatment. In addition, the interviewers probed specifically about the attribute definitions and the understandability of the preference-elicitation exercise. Through these interviews, the study team corroborated patients' ability to complete the DCE questions with the limited information provided for the study attributes. We also verified that the additional information on medication type, indication, and akathisia was considered valuable to some participants. Importantly, interview participants voluntarily shared concerns about the use of antipsychotics and a treatment indication for schizophrenia as they felt taking such

medications would suggest their condition was severe. Finally, the choice question format was confirmed to be consistent with the way the patients thought about treatment choices for themselves. Upon completion of the interviews, the survey was updated based on feedback from the interviewed patients. The final version of the survey can be found in S1 Appendix.

A D-efficient fractional-factorial experimental design was generated in the software package SAS® to populate the attribute levels to be shown in each DCE question. [10] The experimental design provides a way to control the statistical properties of the stimuli in the profiles and minimize the correlations between attributes across DCE questions [11]. Given the number of attributes and attribute levels, we generated a design with 24 questions split into 4 blocks of 6 questions each. We randomly assigned two unique blocks to each respondent, for a total of 12 choice questions [12]. The final design was chosen after using simulations to evaluate the statistical properties of the randomized block assignment and various assumptions about prior preference weights. The order of the blocks of questions and the order of the DCE questions in each block were randomized to address sequence effects.

All activities related to the research were evaluated by the Institutional Review Board (IRB) at a major US research institution (IRB #Pro00107673). Electronic consent was collected from all participants prior to completing the survey instrument.

A total of 501 respondents were recruited through an online consumer panel between October and November, 2021. Potential participants were invited to complete a screener that masked the necessary diagnosis to qualify for the study. Participants were required to self-report a physician diagnosis of MDD and must have failed at least one treatment for depression in the past. Answers to questions about their treatment history were used to verify their report was consistent with diagnosis. Participants who reported having psychotic events were excluded.

## Statistical analysis

Choice data were first evaluated using the internal-consistency measures commonly applied with DCEs [13]. These consistency measures included answers to attribute-comprehension questions and attribute dominance. While we report the incidence of attribute dominance in the sample, we acknowledged that attribute dominance could be a valid expression of preferences, so respondents exhibiting this pattern of choices were retained for the analyses.

We modeled the choice data using logit-based regression models relating the patterns of choices between treatments to the differences in attribute levels [14]. Estimates from logit-based regression models approximate the percentage change in the probability of choice given the attribute levels in the alternatives. For this reason, they also are called preference weights. First, conditional logit models were estimated to evaluate the model specification that would most appropriately characterize the average choice patterns observed in the data. Through these conditional-logit models, we tested the sensitivity of estimates to internal-consistency results and respondents' performance on comprehension questions.

Upon defining the final model specification, we obtained population-level preference estimates that explicitly accounted for preference heterogeneity across respondents by estimating two separate random-parameters logit (RPL) models, one for each of the two information arms in the study (i.e., context vs. no context) [14,15]. Finally, a latent-class (LC) logit model was used to better condition preference heterogeneity on observed characteristics [16], and to identify respondents who were more likely to care about medication indication. We accomplished the latter by running a 3-class model. Preferences in class 1 were required to favor treatments *without* an indication for schizophrenia by constraining the marginal effect of having an indication for schizophrenia to be non-positive. Preferences in class 2 were required to

**Table 2. Model specification by class.**

| Class | Assumed model specification |
|---|---|
| Class 1 – Favors treatments *without* schizophrenia indication | Preference weight for schizophrenia indications was constrained to be non-positive |
| Class 2 – Favors treatments *with* schizophrenia indication | Preference weight for schizophrenia indications was constrained to be non-negative |
| Class 3 – Unconstrained class | All parameters were estimated freely |

favor treatments *with* an indication for schizophrenia by constraining the marginal effect of treating *MDD and schizophrenia* to be non-negative. Preference estimates in class 3 were obtained without any parameter constraints. Table 2 summarizes the model specification by class.

Membership probabilities to class 1 and class 2 reflect the expected number of patients who would find treatment indication to be important. We identified associations between respondent characteristics and class membership using a set of individual-specific covariates to help explain class-membership probabilities. We also used the preference weights from the latent classes to calculate minimum-acceptable treatment effectiveness (MATE) estimates. These values indicate how much chance of efficacy people were willing to give up to avoid treatments presented as antipsychotics or indicated for schizophrenia [17,18].

## Results

Table 3 summarizes characteristics of respondents for the overall sample, as well as separately for those who received additional context information and those who did not. We found no statistically significant differences between respondents who received the additional context information and those who did not. Overall, over three-quarters of the sample was female (76.8%). On average, respondents were 55.7 years of age (SD = 13.1). Close to a third of respondents had a bachelor's degree or higher level of education (31.2%). Also, over half the sample was either retired (31.7%) or unable to work or on disability (21.6%). Finally, a vast majority of the respondents reported feeling at least moderate depression symptoms over the past two weeks (89.4%).

Internal-consistency results suggest data are of good quality with less than 30 respondents (n = 27, 5.4% of the overall sample) exhibiting a choice pattern consistent with response non-variation or taking less than 5 minutes to complete the survey. These 27 respondents were dropped from further analyses. Full internal-consistency results, including conditional-logit sensitivity tests evaluating the impact of these consistency failures, can be found in S2 Appendix.

### DCE results

Fig 2 presents the preference weights derived from the RPL models for the two information arm groups (i.e., context vs no context), normalized relative to the importance of treatment efficacy [19]. This fixes the highest and lowest preference weights for treatment efficacy to be the same across groups. Preference weights represent log-odds indicating whether a specific attribute level increases or decreases the probability of choice relative to another level in the same attribute [14]. As expected, better clinical outcomes were associated with higher preference weights, indicating that respondents were more likely to prefer treatments with those outcomes, all else equal.

The normalized preference weights indicate that respondents in the two groups had different preferences for weight gain and akathisia (marginally) relative to efficacy. We noted that

**Table 3. Survey responses by randomization of contextualization of treatment type.**

| | Statistic or Category | Overall (N = 501) | Context (n = 250) | No Context (n = 251) | P-value |
|---|---|---|---|---|---|
| **All respondents** | | | | | |
| Age in years | n | 501 | 250 | 251 | |
| | Mean (SD) | 55.7 (13.1) | 55.6 (13.2) | 55.9 (12.9) | 0.787 |
| Which of these descriptions most closely describes the worst depression symptoms you have ever experienced? | n | 501 | 250 | 251 | |
| | Very Severe | 119 (23.8%) | 56 (22.4%) | 63 (25.1%) | 0.777 |
| | Severe | 211 (42.1%) | 107 (42.8%) | 104 (41.4%) | |
| | Moderate | 171 (34.1%) | 87 (34.8%) | 84 (33.5%) | |
| Which description above most closely describes your depression symptoms over the past 2 weeks? | n | 501 | 250 | 251 | |
| | Very Severe | 40 (8.0%) | 18 (7.2%) | 22 (8.8%) | 0.220 |
| | Severe | 138 (27.5%) | 60 (24.0%) | 78 (31.1%) | |
| | Moderate | 265 (52.9%) | 143 (57.2%) | 122 (48.6%) | |
| | None of the above | 58 (11.6%) | 29 (11.6%) | 29 (11.6%) | |
| What is your gender? | n | 501 | 250 | 251 | |
| | Male | 115 (23.0%) | 57 (22.8%) | 58 (23.1%) | 0.604 |
| | Female | 385 (76.8%) | 192 (76.8%) | 193 (76.9%) | |
| | Other or prefer not to say | 1 (0.2%) | 1 (0.4%) | 0 | |
| Which of the following describes your ethnicity? (Check only one answer.) | n | 501 | 250 | 251 | |
| | Hispanic, Latino or Spanish | 22 (4.4%) | 12 (4.8%) | 10 (4.0%) | 0.656 |
| | Not Hispanic, Latino or Spanish | 479 (95.6%) | 238 (95.2%) | 241 (96.0%) | |
| What is the highest level of education you have completed? (Check only one answer.) | n | 501 | 250 | 251 | |
| | Less than high school | 2 (0.4%) | 2 (0.8%) | 0 | 0.521 |
| | Some high school | 6 (1.2%) | 3 (1.2%) | 3 (1.2%) | |
| | High school or equivalent (such as GED) | 98 (19.6%) | 52 (20.8%) | 46 (18.3%) | |
| | Some college but no degree | 121 (24.2%) | 56 (22.4%) | 65 (25.9%) | |
| | Technical school | 48 (9.6%) | 30 (12.0%) | 18 (7.2%) | |
| | Associate's degree or 2-year college degree | 70 (14.0%) | 32 (12.8%) | 38 (15.1%) | |
| | 4-year college degree (such as BA, BS) | 82 (16.4%) | 38 (15.2%) | 44 (17.5%) | |
| | Some graduate school but no degree | 7 (1.4%) | 4 (1.6%) | 3 (1.2%) | |
| | Graduate or professional degree (such as MBA, MS, MA, MD, PhD) | 67 (13.4%) | 33 (13.2%) | 34 (13.5%) | |
| Please indicate whether you are currently: (Check all that apply) * | n | 501 | 250 | 251 | |
| | Employed with hourly pay full time | 83 (16.6%) | 42 (16.8%) | 41 (16.3%) | 0.889 |
| | Employed with salary full time | 19 (3.8%) | 8 (3.2%) | 11 (4.4%) | 0.488 |
| | Employed with hourly pay part time | 49 (9.8%) | 25 (10.0%) | 24 (9.6%) | 0.869 |
| | Employed with salary part time | 8 (1.6%) | 5 (2.0%) | 3 (1.2%) | 0.504[b] |
| | Self-employed | 20 (4.0%) | 10 (4.0%) | 10 (4.0%) | 0.993 |
| | Homemaker | 38 (7.6%) | 23 (9.2%) | 15 (6.0%) | 0.173 |
| | Student | 9 (1.8%) | 2 (0.8%) | 7 (2.8%) | 0.176[b] |
| | Retired | 159 (31.7%) | 79 (31.6%) | 80 (31.9%) | 0.948 |
| | Volunteer work | 3 (0.6%) | 1 (0.4%) | 2 (0.8%) | 1.000[b] |
| | Other | 3 (0.6%) | 1 (0.4%) | 2 (0.8%) | 1.000[b] |
| | Not working but looking for a job | 25 (5.0%) | 12 (4.8%) | 13 (5.2%) | 0.845 |

(*Continued*)

**Table 3.** (Continued）

|  | Statistic or Category | Overall (N = 501) | Context (n = 250) | No Context (n = 251) | P-value |
|---|---|---|---|---|---|
|  | Not working and NOT looking for a job | 20 (4.0%) | 10 (4.0%) | 10 (4.0%) | 0.993 |
|  | Unable to work or on disability | 108 (21.6%) | 54 (21.6%) | 54 (21.5%) | 0.981 |

Cm = Centimeters; Lbs = Pounds; Kg/m² = Kilograms per square meters

[a] T-test computed using methods for unequal variance

[b] P-value calculated using Fisher's exact test. Note 1: Percentages do not include missing responses in the denominator. Note 2: P-values are computed using the Pearson chi-square test for categorical variables or the Student's t-test for continuous variables unless otherwise noted.

*Totals for this question may add to more than 100% given that respondents were allowed to select more than one answer.

the additional context information changed the acceptance rate of atypical antipsychotics in a small positive way. The context information was also associated with a greater chance of indifference about the treatment indications.

For the LC logit model, we pooled respondents from the two context arms and allow classification of respondents based on consistency with each of the three preference phenotypes described before (i.e., class 1—preference to avoid schizophrenia indication, class 2—preference to avoid MDD only indication, and class 3 –open to all preference patterns. Figs 3–5 present the estimated preference weights for each of these classes, normalized so the most important attribute—the one showing the greatest change in preference weights—had an absolute effect of 10 units. The expected membership probability for the classes were 63.43% (58.5% - 68.4%), 20.97% (17.1% - 24.9%), and 15.59% (11.7% - 19.5%) for classes 1, 2, and 3, respectively.

Class 1 showed a significant preference for medications indicated only for the treatment of MDD versus a medication indicated for the treatment of *MDD and schizophrenia*. Meanwhile,

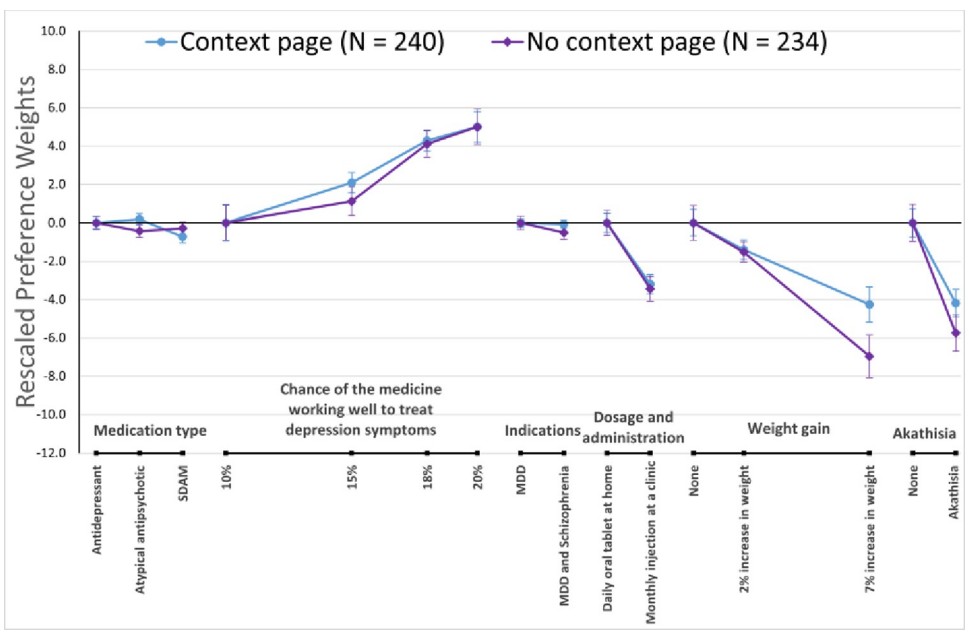

**Fig 2. Rescaled preference weights by information treatment (context vs no context).** MDD = Major Depressive Disorder; SDAM = Serotonin-Dopamine Activity Modulator.

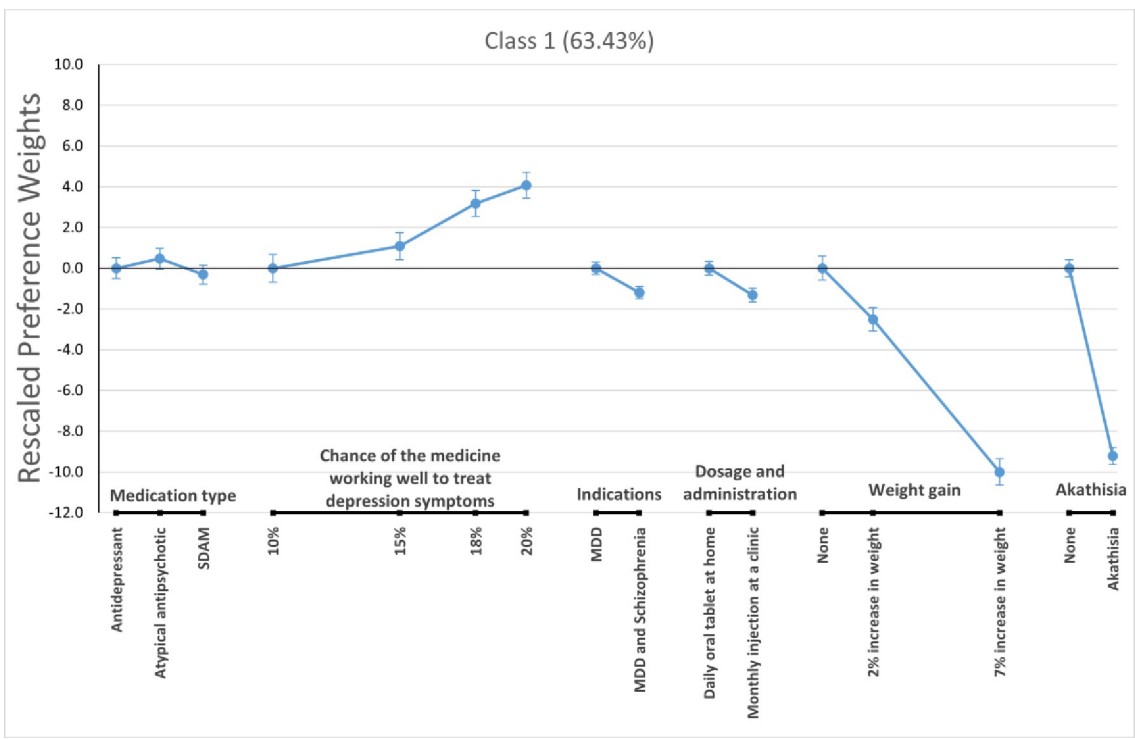

**Fig 3. Rescaled preference weights for latent class 1.** MDD = Major Depressive Disorder; SDAM = Serotonin-Dopamine Activity Modulator.

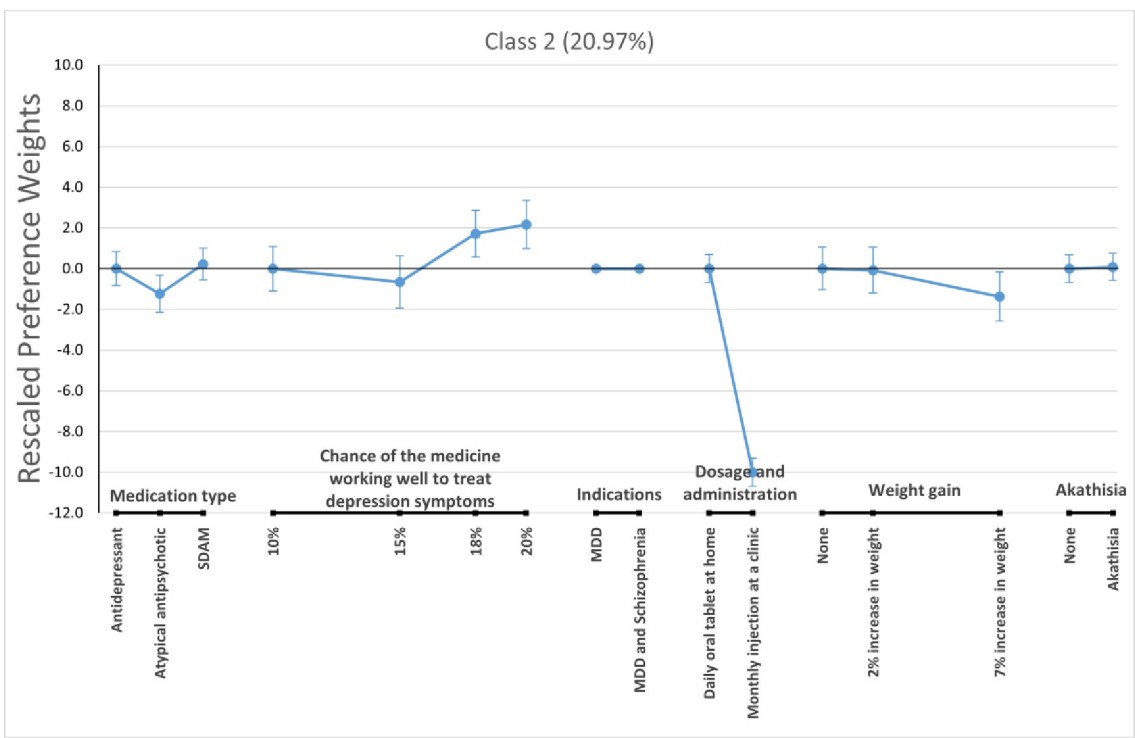

**Fig 4. Rescaled preference weights for latent class 2.** MDD = Major Depressive Disorder; SDAM = Serotonin-Dopamine Activity Modulator.

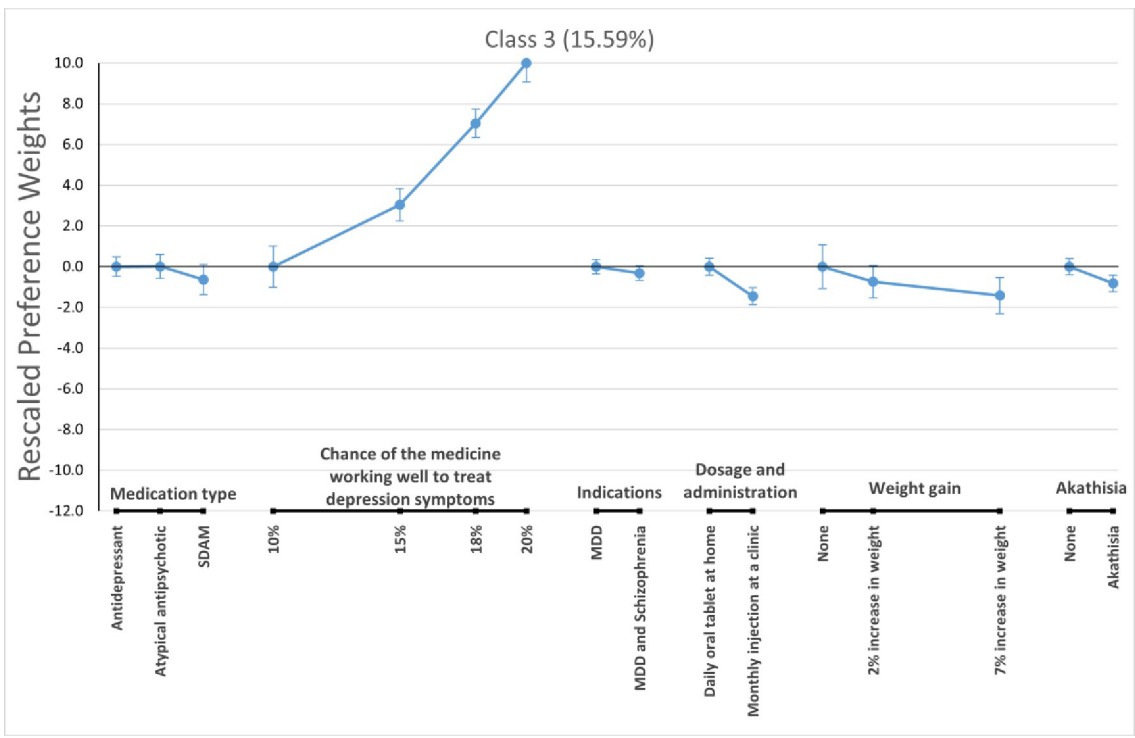

**Fig 5. Rescaled preference weights for latent class 3.** MDD = Major Depressive Disorder; SDAM = Serotonin-Dopamine Activity Modulator.

class 2 and class 3 had flat preferences for treatment indication, suggesting respondents did not consistently prefer treatments that were indicated for both MDD and schizophrenia, but were indifferent between treatment indications. We also found that respondents who were likely to be in class 2 registered strong preferences for daily oral tablets taken at home versus monthly injections at a clinic. Members of class 3 preferred medications that were more effective in reducing depression symptoms. This can be seen by the relatively large increases in preference weights for that attribute as efficacy expectations grew.

Table 4 presents the covariate probabilities characterizing the distribution of each covariate level across the three classes. Recent history of depression and whether the respondents clicked to obtain more information about akathisia were significantly correlated with class assignment. Respondents who reported having more serious depression symptoms in the last 2 weeks were more likely to be found in class 1, the same was true of respondents who clicked for additional information on akathisia. Finally, we found that respondents with higher educational attainment were positively associated with being in class 2, and negatively associated with being in class 1.

## Minimum-acceptable treatment efficacy (MATE)

Table 5 reports MATE estimates for select attribute changes for each latent class. Respondents with a high probability of membership in the first latent class would accept a reduction in the chance of efficacy from 20% to 17.6% if they could have a medication not indicated for the treatment of schizophrenia as well as MDD. The same respondents would have accepted a reduction from 20% to 19.3% to take an antidepressant in lieu of a treatment labeled as a serotonin-dopamine activity modulator. Taking an antidepressant in lieu of a treatment labeled as

**Table 4. Class membership probability by covariate.**

| Covariates | Class1 | Class2 | Class3 | p-value |
|---|---|---|---|---|
| Overall class membership | 0.634 | 0.210 | 0.156 | |
| Depression in the past two weeks | | | | 0.041 |
| Very severe | 0.714 | 0.192 | 0.094 | |
| Severe | 0.679 | 0.178 | 0.143 | |
| Moderate | 0.575 | 0.258 | 0.167 | |
| None of the above | 0.761 | 0.071 | 0.168 | |
| Clicked to learn more about akathisia | | | | 0.007 |
| Did not click | 0.591 | 0.234 | 0.175 | |
| Clicked | 0.790 | 0.121 | 0.089 | |
| Age ranges | | | | 0.190 |
| 21–43 | 0.685 | 0.144 | 0.171 | |
| 44–54 | 0.636 | 0.201 | 0.163 | |
| 55–60 | 0.579 | 0.242 | 0.178 | |
| 61–68 | 0.631 | 0.247 | 0.122 | |
| 69–80 | 0.635 | 0.217 | 0.148 | |
| Education | | | | 0.051 |
| More than high school | 0.667 | 0.184 | 0.150 | |
| High school or less | 0.514 | 0.306 | 0.180 | |
| Years since diagnosis | | | | 0.430 |
| Less than 2 years ago | 0.460 | 0.353 | 0.187 | |
| Between 2 and 5 years ago | 0.614 | 0.159 | 0.227 | |
| Between 5 and 10 years ago | 0.585 | 0.233 | 0.182 | |
| Between 10 and 15 years ago | 0.710 | 0.219 | 0.071 | |
| More than 15 years ago | 0.632 | 0.202 | 0.166 | |
| Treatment history | | | | 0.220 |
| Had a treatment work well in the past | 0.542 | 0.284 | 0.174 | |
| Did not have a treatment work well | 0.657 | 0.192 | 0.151 | |
| Medication history | | | | 0.340 |
| Had a doctor suggest AA | 0.662 | 0.202 | 0.137 | |
| Have not had a doctor suggest AA | 0.597 | 0.221 | 0.182 | |
| Employment | | | | 0.230 |
| Employed part-time or full-time | 0.617 | 0.237 | 0.146 | |
| Other | 0.670 | 0.153 | 0.177 | |

an atypical antipsychotic had a negative value, indicating they would require an increase in efficacy to offset the negative perception of that treatment.

## Discussion

Our study objective was to evaluate whether self-stigmatization was present among patients with MDD when the treatments they are offered are also approved for stigmatized mental-health conditions. We measured the degree to which this avoidance lead to acceptance of reduced treatment efficacy. Additionally, we assessed the impact of adding minimal contextualization of the self-stigmatizing aspects of treatments. Our intention was not to evaluate the right language to discuss treatments with patients, but to evaluate the potential of communication as a tool to counteract self-stigmatization.

**Table 5. MATEs for treatment indication and type.**

| | Improvement | MATE (95% CI)* |
|---|---|---|
| Class 1 | From MDD and schizophrenia to MDD | 17.6% (16.1%, 18.9%) |
| | From SDAM to antidepressant | 19.3% (17.0%, 21.4%) |
| | From atypical antipsychotic to antidepressant** | - |
| Class 2 | From MDD and schizophrenia to MDD*** | - |
| | From SDAM to antidepressant** | - |
| | From atypical antipsychotic to antidepressant | 17.3% (8.3%, 22.4%) |
| Class 3 | From MDD and schizophrenia to MDD | 19.8% (19.3%, 20.3%) |
| | From SDAM to antidepressant | 19.6% (18.8%, 20.3%) |
| | From atypical antipsychotic to antidepressant** | - |

MDD = Major Depressive Disorder; SDAM = Serotonin-Dopamine Activity Modulator.

*From 20% efficacy

**These changes had negative MATEs and were not calculated for the table

*** Class-2 was indifferent to indications.

We found that a product indication can offer important information about treatments and can influence patients' perceptions and their preferences between treatment options. Specifically, we observed that product-label information such as treatment indication can impact patients' preferences for medications to treat MDD. This effect is potentially related to the issue of self-stigmatization. While the effect is relatively small, we found evidence that it is likely pervasive.

A majority of patients were willing to accept statistically significant reductions in potential treatment efficacy to avoid a medication indicated for the treatment of schizophrenia. While the acceptable reductions in efficacy are small, they point to the possibility that people who experience self-stigmatization may be limiting their ability to experience relief from MDD symptoms. Thus, results provide supportive evidence to the idea that self-stigmatization is a real problem among patients with MDD.

Importantly, our results show that psychoeducation can play a key role in preventing stigma in most patients with MDD. We found evidence that additional information on the use of atypical antipsychotics to treat MDD may increase patients' willingness to accept a medication classified as an atypical antipsychotic and reduce the impact of an indication for schizophrenia. Specifically, information regarding treatment efficacy was found to influence willingness to take a medication. Interestingly, the additional context also reduced the negative impact of treatment side effects, like weight gain and akathisia. This highlights the importance of discussions between patients and physicians to address self-stigmatization when treating MDD. Shared decision-making (SDM), in which all treatment options are explored and discussed, is recommended for difficult-to-treat depression [20]. SDM is a systematic process in which providers share information about different treatments, patients are encouraged to discuss their experiences, history, preferences, values, and cultural beliefs, and patients and providers evaluate the pros and cons of each option based on preferences, values, and cultural beliefs to arrive at the best treatment choice. Using SDM may help a patient understand the benefits and risks of treatment choices, and may assist the provider in understanding barriers to use, including self-stigmatization [21].

In our application, simply highlighting the fact that taking antipsychotics does not require experiencing psychotic episodes reduced avoidance of these treatments. Future research should consider measuring how SDM affect actual patient treatment-taking behavior. This

information could help quantify real-world implications of self-stigmatization among MDD patients and support the development of strategies that reduce the avoidance of potentially effective treatments.

Some important limitations of our work are also worth highlighting. One limitation is that the elicited choices did not carry the same consequences as those made in real-world scenarios. However, we setup the choice questions in a way that preference-revealing answers were encouraged. Also, to some degree, our DCE question design included more information than the limited details given on product inserts (i.e., patient-specific treatment efficacy) in order to generate meaningful preference weights. Nevertheless, we focused on limiting the information to the extent possible based on what patients would be expected to find in a product insert. Also, the information treatment provided through the context arm was very basic. Additional details or different communication strategies could have yielded a greater impact for the context information. The nature of our study, also did not explicitly allow us to capture any effects associated with the patient-provider relationship that could moderate the effects we identified in our study. Future studies should measure the impact of these relationships on patients' behaviors and the potential stigma generated by the use of antipsychotics to treat MDD. Finally, while we consider the avoidance of adjuvant antipsychotics because of self-stigmatization, there are other potential reasons for this choice behavior. Our interviews with patients during survey development suggest self-stigmatization was indeed a major driver for this avoidance, but qualitative information on this was not directly collected from the participants who completed the online survey.

## Conclusions

The systematic aversion to adjuvant antipsychotics suggests people who experience self-stigmatization may be limiting their ability to experience relief from MDD symptoms. Discussions between patients and physicians to address self-stigmatization when treating MDD can potentially increase patients' willingness to accept a medication classified as an atypical antipsychotic and reduce the impact of an indication for schizophrenia.

## Supporting information

**S1 Appendix. Survey_submitted.**
(DOCX)

**S2 Appendix. Validity checks.**
(DOCX)

## Acknowledgments

The study team would like to acknowledge the support of Jui-Chen Yang during the analysis of the study data.

## Author Contributions

**Conceptualization:** Juan Marcos Gonzalez Sepulveda, Michael Townsend, Heidi C. Waters, Maalak Brubaker, Reed Johnson.

**Data curation:** Juan Marcos Gonzalez Sepulveda, Matthew Wallace, Reed Johnson.

**Formal analysis:** Juan Marcos Gonzalez Sepulveda, Michael Townsend, Heidi C. Waters, Maalak Brubaker, Matthew Wallace, Reed Johnson.

**Funding acquisition:** Juan Marcos Gonzalez Sepulveda.

**Writing – original draft:** Juan Marcos Gonzalez Sepulveda, Matthew Wallace, Reed Johnson.

**Writing – review & editing:** Juan Marcos Gonzalez Sepulveda, Michael Townsend, Heidi C. Waters, Maalak Brubaker, Matthew Wallace, Reed Johnson.

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
