## [Decision Letter · Decision Letter 0]

29 Mar 2024

PONE-D-24-07744Self-stigmatization and treatment preferences: Measuring the impact of treatment labels on choices for depression medicationsPLOS ONE

Dear Dr. Gonzalez Sepulveda,

Thank you for submitting your manuscript to PLOS ONE. After careful consideration, we feel that it has merit but does not fully meet PLOS ONE’s publication criteria as it currently stands. Therefore, we invite you to submit a revised version of the manuscript that addresses the points raised during the review process.

**ACADEMIC EDITOR: **

**Please review edits as recommended by the reviewer and respond to your comments. Depending on your responses the paper will be further reviewed for acceptance. **

We look forward to receiving your revised manuscript.

Kind regards,

Souparno Mitra, M.D.

Academic Editor

PLOS ONE

“I have read the journal's policy and the authors of this manuscript have the following competing interests: HCW and MB are employees of Otsuka Pharmaceutical Development & Commercialization, Inc. JMGS, RJ, and MW are employees of Duke Clinical Research Institute, which received funding to conduct this study. MT received compensation from Otsuka Pharmaceutical Development & Commercialization, Inc. for his review of the protocol and interpretation of the data for this study.”

Reviewers' comments:

Reviewer's Responses to Questions

**Comments to the Author**

1. Is the manuscript technically sound, and do the data support the conclusions?

Reviewer #1: Yes

Reviewer #2: Yes

2. Has the statistical analysis been performed appropriately and rigorously? 

Reviewer #1: Yes

Reviewer #2: Yes

3. Have the authors made all data underlying the findings in their manuscript fully available?

Reviewer #1: Yes

Reviewer #2: Yes

4. Is the manuscript presented in an intelligible fashion and written in standard English?

Reviewer #1: Yes

Reviewer #2: Yes

5. Review Comments to the Author

Reviewer #1: It is very practical and well thought out manuscript. All parts of the manuscript are easily understood and supported by the statistical and clinically relevant design. Table and figures are simple to comprehend.

Reviewer #2: Dear Editor,

I have had the honor of reviewing the article titled “Self-stigmatization and treatment preferences: Measuring the impact of treatment labels on choices for depression medications” submitted to PLOS One for peer review.

Below is the review:

Under Introduction:

“it is unknown whether self-stigmatization is exacerbated by association with conditions no experienced by the patients” Recommend reconstructing the sentence for clarity: Like “it remains uncertain whether associations with conditions not directly experienced by the patients exacerbate self-stigmatization.”

Under Methods:

“To avoid inducing a reaction from participants in response to these two attributes beyond what would be triggered by the product label” Recommend reconstructing the sentence for clarity: Like “To prevent eliciting a reaction from participants beyond those typically triggered by the product label”

Recommend only using attributes or characteristics and not interchanging them.

Define: akathisia and acronyms: SAS

Recommend using voluntarily instead of spontaneously and taking out the extra they in “Importantly, interview participants spontaneously shared concerns about the use of antipsychotics and a treatment indication for schizophrenia as they felt taking such medications would suggest their condition was severe”

In “The interviews were conducted individually through videoconferencing and followed a

semi-structured think-aloud protocol” Please describe how a semi-structured think-aloud protocol was followed.

“We accomplished the latter by running a 3-class model, where preferences in class 1 were required to favor treatments without an indication for schizophrenia (i.e., constraining the marginal effect of avoiding schizophrenia to be positive), preferences in class 2 were required to favor treatments with an indication for schizophrenia (i.e., constraining the marginal effect of treating MDD and schizophrenia to be positive). Preference estimates in class 3 were obtained without any parameter constraints.” Combine the sentences with a “while”.

Recommend making this a table for clarity as they are referenced frequently under results. In general, the statistical analysis was difficult to follow and understand. Recommend using more description and rewording sentences.

Under Results:

The first paragraph describing the number of participants and description of encounters can go in under

methods.

Use another phrase for “preference weights” or define what is being referenced by preference weights.

Reword for clarity: “We also noted a small, marginally statistically significant positive effect of additional context information on the acceptance of atypical antipsychotics and greater chance of indifference about the treatment indications.”

Under Discussion:

This sentence is not necessary: “While the effect is relatively small, we found evidence that it is likely pervasive”

Under “limitations” section: Other limitations identified are including and limited to not addressing other communication strategies, qualitative information on avoidance of adjuvant antipsychotics because of self-stigmatization was not taken from the participants who completed the online survey, and participants answered on the online survey may not reflect in their real-world decision making.

A suggestion to include is giving specific points that doctors can talk about to their patients that subside the patient’s aversion to adjuvant antipsychotics for MDD symptoms. Consider taking about directions for future research.

Limitations accounted for: consistency check was done where only data of good quality was taken into account and statistical lengths were taken to account for different variances in data.

6. PLOS authors have the option to publish the peer review history of their article (what does this mean?). If published, this will include your full peer review and any attached files.

Reviewer #1: **Yes: **Rajesh Mehta

Reviewer #2: No

---

## [Author Response · Author response to Decision Letter 0]

9 May 2024

COMMENT:

Under Introduction:

“it is unknown whether self-stigmatization is exacerbated by association with conditions no experienced by the patients” Recommend reconstructing the sentence for clarity: Like “it remains uncertain whether associations with conditions not directly experienced by the patients exacerbate self-stigmatization.”

RESPONSE:

We have updated the cited text as suggested by the reviewer.

COMMENT:

Under Methods:

“To avoid inducing a reaction from participants in response to these two attributes beyond what would be triggered by the product label” Recommend reconstructing the sentence for clarity: Like “To prevent eliciting a reaction from participants beyond those typically triggered by the product label”

Recommend only using attributes or characteristics and not interchanging them.

RESPONSE:

We have updated the cited text as suggested by the reviewer.

COMMENT:

Define: akathisia and acronyms: SAS

RESPONSE:

We have added the definition for akathisia immediately after it is first mentioned. 

COMMENT:

Recommend using voluntarily instead of spontaneously and taking out the extra they in “Importantly, interview participants spontaneously shared concerns about the use of antipsychotics and a treatment indication for schizophrenia as they felt taking such medications would suggest their condition was severe”

RESPONSE:

We have updated the cited text as suggested by the reviewer.

COMMENT:

In “The interviews were conducted individually through videoconferencing and followed a

semi-structured think-aloud protocol” Please describe how a semi-structured think-aloud protocol was followed.

RESPONSE:

We have added additional information explaining the process followed as part of the semi-structured think-aloud protocol. The following text has been added to the methods section. 

During the 1-hour individual interviews we asked participants to read the survey out loud and to comment as needed throughout the document on things that seemed unclear or did not match their own experience with the disease or treatment. In addition, the interviewers probed specifically about the attribute definitions and the understandability of the preference-elicitation exercise.

COMMENT:

“We accomplished the latter by running a 3-class model, where preferences in class 1 were required to favor treatments without an indication for schizophrenia (i.e., constraining the marginal effect of avoiding schizophrenia to be positive), preferences in class 2 were required to favor treatments with an indication for schizophrenia (i.e., constraining the marginal effect of treating MDD and schizophrenia to be positive). Preference estimates in class 3 were obtained without any parameter constraints.” Combine the sentences with a “while”. 

Recommend making this a table for clarity as they are referenced frequently under results. In general, the statistical analysis was difficult to follow and understand. Recommend using more description and rewording sentences.

RESPONSE:

Per the reviewer’s suggestion, we updated the text describing the 3 respondent classes in the analysis and added a new table (Table 2) summarizing the information. 

The added text included the following:

We accomplished the latter by running a 3-class model. Preferences in class 1 were required to favor treatments without an indication for schizophrenia by constraining the marginal effect of having an indication for schizophrenia to be non-positive. Preferences in class 2 were required to favor treatments with an indication for schizophrenia by constraining the marginal effect of treating MDD and schizophrenia to be non-negative. Preference estimates in class 3 were obtained without any parameter constraints. Table 2 summarizes the model specification by class.

Table 2. Model specification by class

Class Assumed model specification

Class 1 – Favors treatments without schizophrenia indication Preference weight for schizophrenia indications was constrained to be non-positive

Class 2 – Favors treatments with schizophrenia indication Preference weight for schizophrenia indications was constrained to be non-negative

Class 3 – Unconstrained class All parameters were estimated freely

COMMENT:

Under Results:

The first paragraph describing the number of participants and description of encounters can go in under methods.

RESPONSE:

We have moved the cited paragraph to the end of the method’s section.

COMMENT:

Use another phrase for “preference weights” or define what is being referenced by preference weights.

RESPONSE:

We now have defined the results from a logit-based model of preferences as preference weights. This is because the estimates represent the rate of change in the probability of choices with particular attribute levels. The new text in the analysis section reads as follows.

Estimates from logit-based regression models represent the percentage change in the probability of choice given the attribute levels in the alternatives. For this reason, they also are called preference weights.

COMMENT:

Reword for clarity: “We also noted a small, marginally statistically significant positive effect of additional context information on the acceptance of atypical antipsychotics and greater chance of indifference about the treatment indications.”

RESPONSE:

We have updated the cited text for clarity. The new text reads as follows:

We noted that the additional context information changed the acceptance rate of atypical antipsychotics in a small positive way. The context information was also associated with a greater chance of indifference about the treatment indications.

COMMENT:

Under Discussion:

This sentence is not necessary: “While the effect is relatively small, we found evidence that it is likely pervasive”

RESPONSE:

We would like to maintain this statement as we think it is important to note how widespread the issue of self-stigmatization appears to be in this population based on our results.

COMMENT:

Under “limitations” section: Other limitations identified are including and limited to not addressing other communication strategies, qualitative information on avoidance of adjuvant antipsychotics because of self-stigmatization was not taken from the participants who completed the online survey, and participants answered on the online survey may not reflect in their real-world decision making.

A suggestion to include is giving specific points that doctors can talk about to their patients that subside the patient’s aversion to adjuvant antipsychotics for MDD symptoms. Consider taking about directions for future research.

Limitations accounted for: consistency check was done where only data of good quality was taken into account and statistical lengths were taken to account for different variances in data.

RESPONSE:

In response to the reviewer’s comments we have added the following paragraph to the discussion section. 

In our application, simply highlighting the fact that taking antipsychotics does not require experiencing psychotic episodes reduced avoidance of these treatments. Future research should consider measuring how SDM affect actual patient treatment-taking behavior. This information could help quantify real-world implications of self-stigmatization among MDD patients and support the development of strategies that reduce the avoidance of potentially effective treatments.

---

## [Decision Letter · Decision Letter 1]

19 Jun 2024

PONE-D-24-07744R1Self-stigmatization and treatment preferences: Measuring the impact of treatment labels on choices for depression medicationsPLOS ONE

Dear Dr. Gonzalez Sepulveda,

Thank you for submitting your manuscript to PLOS ONE. After careful consideration, we feel that it has merit but does not fully meet PLOS ONE’s publication criteria as it currently stands. Therefore, we invite you to submit a revised version of the manuscript that addresses the points raised during the review process.

**ACADEMIC EDITOR: **Please see review comments including the ones alluding to including inclusion and exclusion criteria and resubmit for further consideration

We look forward to receiving your revised manuscript.

Kind regards,

Souparno Mitra, M.D.

Academic Editor

PLOS ONE

Reviewers' comments:

Reviewer's Responses to Questions

**Comments to the Author**

1. If the authors have adequately addressed your comments raised in a previous round of review and you feel that this manuscript is now acceptable for publication, you may indicate that here to bypass the “Comments to the Author” section, enter your conflict of interest statement in the “Confidential to Editor” section, and submit your "Accept" recommendation.

Reviewer #2: All comments have been addressed

Reviewer #3: (No Response)

Reviewer #4: All comments have been addressed

2. Is the manuscript technically sound, and do the data support the conclusions?

Reviewer #2: Yes

Reviewer #3: Partly

Reviewer #4: Yes

3. Has the statistical analysis been performed appropriately and rigorously? 

Reviewer #2: Yes

Reviewer #3: I Don't Know

Reviewer #4: Yes

4. Have the authors made all data underlying the findings in their manuscript fully available?

Reviewer #2: Yes

Reviewer #3: No

Reviewer #4: Yes

5. Is the manuscript presented in an intelligible fashion and written in standard English?

Reviewer #2: Yes

Reviewer #3: Yes

Reviewer #4: Yes

6. Review Comments to the Author

Reviewer #2: Thank you for the opportunity to rereview "Self-stigmatization and treatment preferences: Measuring the impact of treatment labels on choices for depression medications". All the requested review comments were adequately addressed.

Reviewer #3: Self-stigmatization and treatment preferences: Measuring the impact of treatment labels on choices for depression medications.

Summary of research

The paper by et al. analyzed the influence of medication product inserts on patients’ perception and their preference between treatment options. 501 Patients with MDD were made to complete DCE survey. Based on the result author suggest, Product-label treatment indication can potentially lead to patient self-stigmatization as shown by patients’ avoidance of treatments that are also used to treat schizophrenia.

Weaknesses:

1) In the study participants were made to read the label and make recommendations. It is important to consider the percentage of patients who read the labels of the medications in detail in real life scenario.

2) Psychoeducation plays a key role in preventing stigma in most patients with MDD. It is usual practice for psychiatrists to explain the risks, benefits, FDA black box warning, other indications of medications prescribed. Psychiatrists usually explain these to the patient before starting new meds and make them understand why specific medication was chosen for them and how it could help them. These discussions usually help in preventing stigma in most cases. I am curious to know in the study sample if a provider explained them the above and answered all questions.

3) Study states following: The additional information was intended to provide more context around the use of atypical antipsychotics to treat MDD mimicking the kind of information a patient could receive from physicians to alleviate problems with self-stigmatization. Respondents who were offered the additional information (context arm) were reminded explicitly that patients need not experience psychoses to benefit from antipsychotics.

---- It is important to note: The therapeutic alliance made between patients and psychiatrist usually makes a great difference in stigma, as patients trust on provider is greater to a researcher explaining them.

4) In discussion it states: Also, to some degree, our DCE question design included more information than the limited details given on product inserts to generate meaningful preference weights.

It is important to know what additional information was added as it changes the validity of the study, and its application to real life scenarios.

Major issues:

1) Inclusion criteria: Specification regarding inclusion criteria is missing. Recommended to write clear inclusion criteria.

Pl refer to PLOS ONE submission guideline: https://journals.plos.org/plosone/s/submission-guidelines#:~:text=In%20the%20text%2C%20cite%20the,not%20include%20citations%20in%20abstracts

5) In methods pl include the sample size.

6) Exclusion criteria needed to be added.

Minor issues:

1) We included two attributes potentially associated with self-stigmatization. First, medication type, which included antidepressants and two different names that could be used to describe the same family of antipsychotics to treat MDD. It is difficult to follow this sentence. Recommend rephrasing

General comments:

While the data from this article is informative. Study methodology is unclear with missing information to reproduce the study. Also, study could be more valid if questionnaire used included psychiatrist/ physician involvement in recommending the medication and discussing with patient

Reviewer #4: This article offers valuable insights into self-stigmatization and how it interacts with treatment and medication compliance. Although some of the conclusions were difficult to grasp, overall, the article successfully identifies the impact of self-stigmatization on the treatment of specific mental health issues using a decent sample size.

7. PLOS authors have the option to publish the peer review history of their article (what does this mean?). If published, this will include your full peer review and any attached files.

Reviewer #2: No

Reviewer #3: **Yes: **Mallikarjuna Bagewadi Ellur

Reviewer #4: No

---

## [Author Response · Author response to Decision Letter 1]

17 Jul 2024

We have edited the manuscript to address the comments from the reviewer. We hope the changes made satisfy the reviewer. Thanks for the opportunity to respond to that feedback.

---

## [Decision Letter · Decision Letter 2]

14 Aug 2024

Self-stigmatization and treatment preferences: Measuring the impact of treatment labels on choices for depression medications

PONE-D-24-07744R2

Dear Dr. Gonzalez Sepulveda,

We’re pleased to inform you that your manuscript has been judged scientifically suitable for publication and will be formally accepted for publication once it meets all outstanding technical requirements.

Kind regards,

Souparno Mitra, M.D.

Academic Editor

PLOS ONE

Additional Editor Comments (optional):

Reviewers' comments:

Reviewer's Responses to Questions

**Comments to the Author**

1. If the authors have adequately addressed your comments raised in a previous round of review and you feel that this manuscript is now acceptable for publication, you may indicate that here to bypass the “Comments to the Author” section, enter your conflict of interest statement in the “Confidential to Editor” section, and submit your "Accept" recommendation.

Reviewer #2: All comments have been addressed

Reviewer #4: All comments have been addressed

2. Is the manuscript technically sound, and do the data support the conclusions?

Reviewer #2: Yes

Reviewer #4: Yes

3. Has the statistical analysis been performed appropriately and rigorously? 

Reviewer #2: Yes

Reviewer #4: Yes

4. Have the authors made all data underlying the findings in their manuscript fully available?

Reviewer #2: Yes

Reviewer #4: Yes

5. Is the manuscript presented in an intelligible fashion and written in standard English?

Reviewer #2: Yes

Reviewer #4: Yes

6. Review Comments to the Author

Reviewer #2: Thank you for the opportunity to rereview "Self-stigmatization and treatment preferences: Measuring the impact of treatment labels on choices for depression medications". All the requested review comments by all the reviewers were adequately addressed. All the suggested recommendations were done and addressed appropriately.

Reviewer #4: The examination underscores the significance of psychoeducation and shared decision-making in addressing self-stigmatization and enhancing the acceptance of treatment among individuals with Major Depressive Disorder (MDD). By ensuring that patients are well-informed and actively engaged in their treatment choices, healthcare providers can effectively confront stigma-related barriers and facilitate more proficient management of depression.

7. PLOS authors have the option to publish the peer review history of their article (what does this mean?). If published, this will include your full peer review and any attached files.

Reviewer #2: No

Reviewer #4: No

---

## [Editor Report · Acceptance letter]

23 Aug 2024

PONE-D-24-07744R2 

PLOS ONE

Dear Dr. Gonzalez Sepulveda, 

I'm pleased to inform you that your manuscript has been deemed suitable for publication in PLOS ONE. Congratulations! Your manuscript is now being handed over to our production team.

Kind regards, 

on behalf of

Dr. Souparno Mitra 

Academic Editor

PLOS ONE